# Cyanotoxins and Other Bioactive Compounds from the Pasteur Cultures of Cyanobacteria (PCC)

**DOI:** 10.3390/toxins15060388

**Published:** 2023-06-09

**Authors:** Muriel Gugger, Anne Boullié, Thierry Laurent

**Affiliations:** Institut Pasteur, Université Paris Cité, Collection of Cyanobacteria, 75015 Paris, France

**Keywords:** cyanobacteria, the Pasteur Cultures of Cyanobacteria, cyanotoxins, natural products

## Abstract

In tribute to the bicentenary of the birth of Louis Pasteur, this report focuses on cyanotoxins, other natural products and bioactive compounds of cyanobacteria, a phylum of Gram-negative bacteria capable of carrying out oxygenic photosynthesis. These microbes have contributed to changes in the geochemistry and the biology of Earth as we know it today. Furthermore, some bloom-forming cyanobacterial species are also well known for their capacity to produce cyanotoxins. This phylum is preserved in live cultures of pure, monoclonal strains in the Pasteur Cultures of Cyanobacteria (PCC) collection. The collection has been used to classify organisms within the Cyanobacteria of the bacterial kingdom and to investigate several characteristics of these bacteria, such as their ultrastructure, gas vacuoles and complementary chromatic adaptation. Thanks to the ease of obtaining genetic and further genomic sequences, the diversity of the PCC strains has made it possible to reveal some main cyanotoxins and to highlight several genetic loci dedicated to completely unknown natural products. It is the multidisciplinary collaboration of microbiologists, biochemists and chemists and the use of the pure strains of this collection that has allowed the study of several biosynthetic pathways from genetic origins to the structures of natural products and, eventually, their bioactivity.

## 1. Introduction

Cyanobacteria represent a monophyletic lineage of Gram-negative oxygenic photosynthetic bacteria [1]. This phylum has inhabited the Earth for 2.8 billion years, contributing to changes in the geochemistry and the biology of the globe [2]. They are ubiquitous and found in diverse ecological niches, from aquatic ecosystems such as lakes, rivers and oceans to deserts, Polar Regions, caves and even in symbiosis with other organisms, such as fungi, to form lichens, for example. Cyanobacteria are also the ancestors of chloroplasts that are found in most plants and algae, while recurrent examples of various cyanobacterial morphotypes are found associated with the leaves or in the roots of plants. Although cyanobacteria are famous for the oxygenation of the Earth [2,3], they are also sadly famous for the toxic blooms they can massively develop in marine and fresh waters all around the globe [4]. The field of cyanotoxins and other cyanobacterial bioactive compounds has greatly extended over the last century, notably through chemical analysis and compound structure elucidation, and due to the availability of enough material for toxicological studies. Despite 1100 cyanobacterial natural products (NPs) discovered by these approaches [5], we still know little about them. In this review, we will first introduce the diversity of the cyanobacterial phylum; secondly, we will present the repository dedicated to this phylum at the Institut Pasteur, and finally, we will discuss how to exploit the cyanobacterial biobank to reveal novel compounds, corresponding pathways, new enzymes and even intriguing chemistry. We will end this overview with examples of bioactivity studies on cyanotoxins and NPs.

## 2. The Phylum Cyanobacteria, Diversity in Terms of Morphology and Genome

The phylum of Cyanobacteria containing all bacteria capable of performing oxygenic photosynthesis [1] presents a wide breadth of habitats and, thus, of ecology, physiology and morphology. Their most simple morphotypes can be unicellular, as single cells, e.g., *Prochlorococcus*, Chisolm et al. (1992), that colonize the oceans [6], or as colonies of single cells embedded in mucilage, e.g., *Microcystis*, Kützing ex Lemmermann (1907), that form toxic blooms in lakes [7]. The colonies of unicellular cyanobacteria can be tightly organized in one layer of arranged cells of *Merispomedia*, Meyen (1839), or can appear as cell aggregations of various sizes surrounded by mucilaginous envelopes such as the tiny *Chroococus*, Nägeli (1849), and the large *Gloeocapsa*, Kützing (1843). More complex morphotypes of single cells in a colony are the *Pleurocapsa*, Thuret in Hauck (1885), and other baeocytous cyanobacteria that proliferate in desert environments. These cyanobacteria are extremely resistant to desiccation; containing baeocytes that will revive a novel colony when the environmental conditions are more favorable for the growth of these bacteria. Filamentous cyanobacteria also have a more organized cell division which is perpendicular to the growing axis of the filament, such as the solitary trichomes found in toxic freshwater blooms such as *Planktothrix*, Gaget et al. (2015) or the several trichomes embedded in a common sheath such as *Hydrocoleum*, Kützing ex Gomont (1892). The filamentous heterocystous cyanobacteria, exemplified by *Byssus flos-aquae*, Linnaeus (1753), are even more complex; this taxa is currently invalid and has been replaced by *Aphanizomenon flos-aquae*, Ralfs ex Bornet and Flahault (1886). This morphotype bears differentiated cells such as akinetes, used to revive a novel filament after harsh conditions, and heterocytes, used to fix atmospheric nitrogen. *Aphanizomenon* is also a toxic bloom-forming cyanobacterium found in lakes and rivers. Finally, the most complex morphotypes, and often larger than other cyanobacterial morphologies, are the filamentous cyanobacteria with differentiated cells (heterocytes and akinetes) and true ramifications, surrounded or not by mucilage and sheaths. The best ubiquitous representative of this cyanobacterial morphotype is *Fischerella* (Bornet and Flahault) Gomont, 1895. This summarizes at a glance the morphological diversity of Cyanobacteria that can be seen with the naked eye or using a light microscope. A glimpse into the morphologies encountered in the cyanobacterial phylum can be found in Figure 1; a more detailed view of the morphological diversity, through a botanical approach with camera lucida drawings, can be found in the book series Süsswasserflora von Mitteleuropa on Cyanoprokaryota [8,9,10]. Moreover, a detailed view of the morphological diversity through a bacteriological approach with photography of the various morphologies and electron microscopy photographs of the ultrastructural arrangement within the cyanobacterial cells can be found in the Bergey’s Manual of Systematic Bacteriology [11].

In response to the morphological diversity within the phylum of Cyanobacteria, their genomes are also extremely diverse. The publicly available genomes present sizes ranging from 1.5 to 15 Mb and a GC content from 30 to 68 %. The smallest genome size corresponds to the picocyanobacterial genus of *Prochlorococcus*, while the largest ones were found in filamentous cyanobacteria with differentiated cells and ramifications. On the contrary, the highest and lowest GC contents were reported in the genus *Synechococcus*, Nägeli (1849), while the filamentous cyanobacteria with differentiated cells and with or without branching presented a GC content centered on approximately 42%. The comparison of the morphologies and ultrastructural data with genomic sequence data did not show any solid groupings corresponding to the different morphotypes; however, a correlation between the ultrastructures and genes coding for cellular inclusions was identified [12].

## 3. Collection of Cyanobacteria and the Pasteur Cultures of Cyanobacteria (PCC)

Several collections of cyanobacteria have been constituted in universities and institutions around the world, as cyanobacterial blooms occur everywhere on Earth. In addition, a few major collections often conserved these bacteria along with algal isolates, protists and other bacteria (for example, NIES-MCC in Japan; UTEX and ATCC in the USA; GCC, NFMC and NCCS in India; CCAP in Scotland, CCALA in Czech Republic; BCCM-ULC in Belgium; NIVA in Norway; UHCC in Finland; SAG in Germany; PCC, PMC, TCC and RCC in France). The collection of cyanobacteria currently present at the Institut Pasteur arrived from the University of California, Berkeley, with Pr. Roger Stanier (1916–1982), who worked on the classification of the so-called blue-green algae at that time and included them among the bacterial classification as the Cyanobacteria [13]. Pr. Stanier led a research team named Microbial Physiology Unit, which focused on various subjects such as pigments, photosynthesis, gas vacuoles, fatty acids and taxonomy, upon of these 150 pure cyanobacterial strains. From the same laboratory, the isolation of a purple cyanobacterium without thylakoids, internal membranes on which the light-harvesting complexes sit, led to the description of the first representative of the basal clade of the phylum Cyanobacteria, *Gloeobacter violaceaus*, Rippka, Waterbury and Cohen-Bazire (1974), strain PCC 7421 [14]. After a short period (1982–1988) during which Dr. Germaine Cohen-Bazire Stanier (1920–2001) was heading this team and working on the ultrastructure of cyanobacteria [15,16], Pr. Nicole Tandeau de Marsac (1944–2020) was named head of a novel team called the Unit of Cyanobacteria from 1988–2009. This team maintained the cyanobacterial collection of the Institut Pasteur, which had greatly expanded since its arrival in France through the effort of several researchers and visitors. She and collaborators worked extensively on a few model PCC strains to thoroughly investigate the phycobilisome and the photoregulation, the cellular differentiation of hormogonium, the gas vesicle genes, and the phosphorylation of the signal transducer PII [17,18,19,20,21]. Pr. Tandeau de Marsac also revealed the complementary chromatic adaptation [22,23,24]. Towards the end of her research career, she majorly focused on the toxic cyanobacterial isolate *Microcystis aeruginosa*, PCC 7806 [7,25,26,27]. In July 2009, Dr. Muriel Gugger was appointed head of the Collection of Cyanobacteria with the mission to maintain, distribute and valorize the Pasteur Cultures of Cyanobacteria collection. The PCC collection is also used as a reference for this phylum and to exemplify the bacterial classification in Bergey’s Manual [11].

Since Stanier’s time, several collaborators and visitors cooperated with Rosemarie Rippka (1944–today), the curator of the PCC collection until June 2009, to carry out research [28,29,30] and, in the meantime, bred up to 800 axenic living PCC strains isolated from all over the world (Figure 2). From the 150 strains that originally arrived with Stanier, 102 are still maintained at the PCC, with the emblematic strain *Synechococcus elongatus*, PCC 6301, isolated in 1956 from the USA, rendered monoclonal and axenic by 1963 and maintained alive since then as well as entered in cryopreservation stocks several times as all other PCC strains [31]. In 1973 and 1989, strong efforts generated about 100 cyanobacterial monoclonal purified isolates incorporated in the collection, but over time, several isolates were lost or stopped growing. The purification of cyanobacteria is a long delicate process that can take several years, as exemplified by the two years required to obtain an axenic monoclonal culture of *Prochlorococcus marinus*, PCC 9511 [32], which vanished in 2013 due to the replacement of the incubator in which it was maintained. The last strains incorporated in the collection also belong to the genus *Synechococcus*; two come from India and are closely related to the PCC 6301 strain but represent another species, and one comes from Singapore described as a transformable strain with a fast-growing capacity under high light [31,33,34,35]. Since 2006, the PCC collection was integrated into the Biological Resource Center of the Institut Pasteur (CRBIP) along with the Collection of Bacteria of the Institut Pasteur (CIP) and the fungal collection (former CMIP) and joined later under the same umbrella of biobank and quality management by the Collection Nationale de Cultures de Microorganismes (CNCM) and the Integrated Collections for Adaptive Research in Biomedicine (ICAReB-Biobank). Today, the PCC collection contains around 800 monoclonal axenic cultures, 600 maintained alive in liquid or solid cultures, and all of them cryopreserved. The maintenance of the collection takes up to 4500 transfers, along with 4500 purity tests per year. While the PCC collection represents a breadth of the cyanobacterial phylum, the study of their genomes of diverse PCC strains revealed a plethora of gene clusters for NPs [3,36].

## 4. Natural Products of Cyanobacteria—From Toxins to Novel Compounds

In the 1870s, a report documented the occurrence of cattle poisonings from Australian lakes [37]. Over 100 years later, dog deaths became a vivid subject connected to cyanotoxins from cyanobacterial developments on the surface or benthos of water ecosystems in North America and Europe, as well as in South Africa and New Zealand [38,39,40,41,42,43,44]. More recently, a human fatality during a renal dialysis treatment in Brazil and a fatality in bald eagles in the southeastern USA were demonstrated to be due to freshwater cyanobacterial occurrences [45,46]. Several wildlife intoxications or deaths have also been associated with cyanotoxins, for example, with flamingos and most probably with elephants in Africa and with fishes in Canada [47,48,49]. The above examples clearly demonstrate that for the last 145 years, the recurrent problem of cyanobacterial blooms in fresh and marine water bodies has presented a threat to animals and humans. Moreover, the cyanotoxins released by bloom-forming cyanobacteria into water bodies used for drinking water create a global public health issue. The World Health Organization has developed guidance values for the most common cyanotoxins present in recreational water and drinking water [50]. In the USA, the simultaneous incident of animal and human disease around one lake with blooms of toxic cyanobacteria has been documented and has called for the development of a proactive relationship between the healthcare system and veterinarians to protect human health [51,52]. In the field of cyanotoxins, the monoclonal and axenic PCC strains have been useful since 1988. For example, the *Microcystis aeruginosa* PCC 7820 was used to monitor the hepatotoxic effect of microcystin-LR on mice and rat liver damages and pulmonary emboli leading to acute toxicities and death [53]. In particular, two strains have made it possible to discover the genetic bases of three cyanotoxins: *Microcystis aeruginosa* PCC 7806 for the discovery of the microcystin biosynthetic gene cluster, and *Kamptonema* sp. PCC 6506 to reveal anatoxin-a and related compounds as well as cylindrospermopsins [54,55,56]. Indeed, a recent bibliographic survey (27th April 2023, in Pubmed: Cyanobacteria AND PCC AND Toxin) reveals about 150 publications dedicated to cyanotoxin discovery, cyanotoxin effects (larvicidal, antifungal, …), toxin-antitoxin and treatment against toxic cyanobacteria based on PCC strains.

Cyanobacteria do not only produce toxins; they contain a real diversity in terms of natural substances. In order to have greater visibility of this diversity, we undertook the sequencing of the genomes of the living axenic strains preserved in the PCC collection. First, we obtained the genomes of 54 PCC strains selected on their morphology, their ecology and their physiology to better represent the breadth of cyanobacterial phylum. Combined with genomic data from 72 publicly available strains, a phylogenetic tree based on 31 genes conserved in Bacteria was constructed to reflect the evolution and relationship between these organisms. In parallel, a systematic analysis of gene clusters coding for NPs and toxins was undertaken on this dataset with the search for ribosomally synthesized and post-translationally modified peptides (RiPPs), non-ribosomal peptides synthetases (NRPS) and polyketide synthases (PKSs) [3]. This analysis showed the presence of these three classes of metabolites, with *cis* and *trans* AT-PKS, peptides from both ribosomal and non-ribosomal pathways, and terpenes throughout the phylum represented by these 126 genomes. In line with this work, a focus on the NRPS and PKS in the same dataset made it possible to highlight 452 biosynthetic gene clusters (BGCs) distributed into 286 cluster families based on the similarity of the modules of the NRPS and PKS and the length of the regions compared [36]. Interestingly, one-fourth of the BGCs were hybrids of NRPS and PKS, mostly distributed in the late branches of the cyanobacterial phylogeny, whereas the early branches contained mainly PKS. In addition, 80% of these cluster families did not correspond to any known NPs, giving an idea of the scope of the investigative work, which could be devolved by chemists and biochemists.

Based on the above findings and the availability of cyanobacterial strains potentially producing unknown NPs in the PCC collection, collaborations with various chemist and biochemist colleagues helped reveal the NPs derived from these unattributed BGCs. For the BGCs smaller than 15 kb, a cloning strategy performed in a heterologous host was more straightforward to discover compounds of interest and to find them further in the cyanobacterium of interest, such as the schizokinen-like siderophore of *Leptolyngbya* sp., PCC 7376 [57]. However, most of the NRPS-PKS BGCs were larger than 20 kb, and they could not be produced with this approach. For the investigations of these large BGCs, we sometimes cultivated liters of biomass of pure cyanobacterium, potentially producing the desired metabolites in our laboratory conditions. Through numerous collaborations, we have discovered more than 20 novel NPs and/or their BGC, thanks to the cyanobacteria in the PCC collection and the collaborators, as well as other researchers with PCC strains (Table 1).

Finally, in the investigation course of these NPs by a multidisciplinary consortium of chemists, biochemists and microbiologists, we were fascinated to find novel enzymes and unknown chemistry from the metabolisms of the cyanobacterial strains of the PCC, examples of which are discussed hereafter. First, the search for the proteusins of the cyanobacteria and the radical *S*-adenosyl methionine epimerase (rSAM) of this pathway revealed regioselective *D*-configured amino acids into peptidic NPs. For this, an ingenious methodology was developed to understand how these rSAMs work to irreversibly insert multiple *D*-amino acids in the peptides from the strains *Kamptonema* sp. PCC 6506, *Pleurocapsa* sp. PCC 7319, and *Anabaena variabilis* ATCC 29413 [58,59]. Secondly, the same talented chemists working with the genomic data of *Pleurocapsa* sp. PCC 7319 discovered non-canonical protein splicing via a post-translational excision of a tyramine equivalent, leading to an α-keto-β-amino amide [60]. Third, in the RiPPs family of chemically diverse cyanobactins, the BGC is highly conserved and thus, the genes coding the enzymes of these pathways are named consistently from A to F. Nevertheless, several F enzymes enlarged the prenyltransferase family, with two of these enzymes in the muscoride pathway acting differently by introducing a regioselective prenylation on the amino acid termini of the produced linear cyanobactin in *Nostoc* sp., PCC 7906 [61], or with another prenyltransferase which places a forward-prenyl on a threonine residue in the cyclic cyanobactin tolypamide of *Tolypothrix* sp. PCC 7601 [62]. Finally, the strain *Lyngbya* sp. PCC 8601 and two other cyanobacterial strains were used to uncover a suite of post-translational modifying bacterial enzymes that install single or multiple strained cyclophane macrocycles. As the cyclophane natural products are found in fungi, plants and bacteria; this enzyme family is widely distributed in nature [63].

## 5. From Molecules to Bioactivity

From the discovery of new molecules or their genetic heritage to the knowledge of their activity, the path is not straightforward. Initially, the origin of a toxic event was sought before finding the cyanotoxin responsible for it. Several reviews described the potential of cyanobacterial compounds to become drug products such as anticancer agents and antibiotics, for example [64,65,66,67]. Chemists have found more than 1100 NPs from cyanobacteria, but less than 20% of them are associated with a biosynthesis pathway [5]. In these chemical studies, it has often been attempted to find bioactivity associated with it by means of conventional screening techniques, in particular, to find a therapeutic potential. However, almost half of the compounds were not tested or detected in any bioactivity assay. More recently, a review on the bioactivity of NPs of cyanobacteria found 1630 unique molecules, classified into 260 families of metabolites [68]. Importantly, most of the compounds were not tested for their bioactivities. This is because bioactivity testing requires different knowledge and specialty than that needed to discover the compounds or the genetic data that encodes them. In addition, as the characterisation of a compound will often require the extraction and collection of grams of it from the biomass of the producing organism, it is often for lack of material that the bioactivity test cannot be carried out or confirmed.

**Table 1 toxins-15-00388-t001:** Natural products (NPs) and their biosynthetic gene clusters (BGC) discovered based on the monoclonal and axenic strains of the collection PCC.

NPs or Their BGCs	Type of NPs	Gene Cluster	Strain	Reference
Microcyclamide	RiPPs, cyanobactins	*mca*, 13 kb	*Microcystis aeruginosa* PCC 7806	[26]
Viridisamide	RiPPs, 1st linear cyanobactins	Variation from *pat* *	*Oscillatoria viridis* PCC 7112	[69]
Aeruginosamide B and C3			*Microcystis aeruginosa* PCC 9432	
Muscoride	RiPPs, linear cyanobactin	*mus*, 12.7 kb	12 *Nostoc* strains, in which 6 PCC strains	[61]
Tolypamide	RiPPs, cyanobactins	*tol*, 10.4 kb	*Tolypothrix* sp. PCC 7601	[62]
Geosmin	RiPPs, sesquiterpene	*geosmin synthase*	*Nostoc* spp. *PCC 7310 and PCC 7120*, *Kamptonema* sp. PCC 6506	[70,71]
Merosterol A and B + isomer	RiPPs, meroterpene	*mst*, 29 kb	*Scytonema* sp. PCC 10023	[72]
Cyclophanes	RiPPs, cyclopetide alkaloids	*lsc*, 2.6 kb	*Lyngbya* sp. PCC 8106 and various other strains	[63]
Landornamides	RiPPs, proteusins	*osp*, 12 kb	*Kamptonema* sp. PCC 6506 and 6 other *Kamptonema* PCC strains	[73]
Kamptornamide	RiPPs, 1st ribosomal fatty-acylated lipo- petides, selidamides	*ksp*, 6.3 kb	*Kamptonema* sp. PCC 6506 and *Nostoc punctiforme* PCC 73102	[74]
Microguanidine amide Aeruginoguanidine BGC	NRPS	*agd*, 34 kb	11 *Microcystis*, in which 7 PCC strains	[75]
Hassallidin E	NRPS	*has*, 48 kb	*Planktothrix serta* PCC 8927	[76]
Cyanopeptolin	NRPS	*oci* *, 31.5 kb	*Microcystis aeruginosa* PCC 7806 *Scytonema hofmanni* PCC 7110	[77,78]
Scyptolins	NRPS		*Scytonema hofmanni* PCC 7110	[79]
Anatoxin-a and dihydroanatoxin	PKS, alkaloid	*ana*, 20 kb	*Kamptonema* sp. PCC 6506, *Cylindrospermum* sp. PCC 7417 and 13 other PCC strains	[36,55]
Cylindrospermopsins	PKS, alkaloid	*cyr*, 42 kb	*Kamptonema* sp. PCC 6506	[56]
Luminaolide B	Trans AT-PKS	*lum*, 99 kb	*Planktothrix paucivesiculata* PCC 9631	[80,81]
Tolytoxin BGC		*tto*, >100 kb	*Scytonema* sp. PCC 10023	
Tolytoxin, Scytophycin		*tto*, 92.8 kb	*Planktothrix* sp. PCC 11201	
Leptolyngbyalide	Trans AT-PKS	*lept*, 96.7 kb	*Leptolyngbya* sp. PCC 7375	[82]
Alkene and alkanes	PKS, Hydrocarbon	*ols* 10 kb	*Synechococcus* sp. PCC 7002 16 unicellular PCC strains	[83,84,85]
Heterocyte glycolipids	PKS, polyunsaturated fatty acid	CF1 *	*Nostoc* sp. PCC 7120, 18 Nostocales strains and *Microchaete* sp. PCC 7126	[86,87,88]
Microcystin	NRPS-PKS Hybrid	*mcy*, 55 kb	*Microcystis aeruginosa* PCC 7806, *Fischerella* sp. PCC 9339	[36,54]
Nostopeptolide	NRPS-PKS Hybrid	pks2, 62.7 kb	*Nostoc puctiforme* ATCC29133/PCC 73102	[89,90]
Aranazoles	NRPS-PKS Hybrid	*arz*, 43 kb	*Fischerella* sp. PCC 9339	[91]

* Indicates: *pat* from *Prochoron*, a cyanobacterial symbiont of a tunicate [92]; *oci* from diverse cyanobacteria [93]; and CF1 correspond to the cluster in *Nostoc* sp. PCC 7120 [36,88].

The study of biological activity can be directed to help human interest, for a pharmacological application or a biotechnological development. Tolytoxin has the potential to be used for therapeutic application, as this cyanobacterial macrolide inhibits actin filament dynamics and was proposed as a potential anti-cancer drug [94,95,96]. However, this molecule was also proved to be extremely toxic at nM concentrations and to induce cell death [94,97]. As we recently described tolytoxin producers and the tolytoxin biosynthetic gene cluster from PCC pure strains [80], we found several other PCC strains capable of producing this molecule [81,98]. We revisited the activity of tolytoxin in human cells from neuronal and epithelial origins with the goal of reducing disease transmission by tunneling nanotubes mainly constituting of actin [98]. In this experiment, with the two strains we used, we noticed a strong decrease in tolytoxin dose needed (3 and 15 nM) to obtain an inhibitory effect without setting off the toxic side effects previously observed in cells. During the isolation of pure tolytoxin from *Planktothrix serta* PCC 8926 and *Scytonema* sp. PCC 10023, we noticed that a fatty acid was extracted along with the tolytoxin almost until the end of our extraction procedure. To perform this experiment and extract enough purified tolytoxin, we worked in a chemist’s laboratory, thanks to J. Piel’s team and R. Ueoka in particular, without whom we would have missed this trace of contamination. For the activity of tolytoxin, we collaborated with specialists in the cells and nanotubes to be tested, thanks to C Zurzolo’s team and A. Dilsizoglu-Senol in particular [98]. In addition, the odorous volatile compound geosmin is also of concern for human health with its biological activity. Using the geosmin-producing strain *Kamptonema* sp. PCC 6506, and the non-producing strain *Leptolyngbya* sp. PCC 8913 isolated from a lake colonized by mosquitoes in the south of France [99], we collaborated with researchers working on insect olfaction at Lund University (Sweden) to reveal the attraction of the *Aedes aegypti* mosquito for this compound and check if this odour is an indicator of egg-laying site for this insect [70]. 

The study of biological activity can also be investigated to learn more about its need for producing cyanobacteria. Two clear examples of useful NPs for producing cyanobacteria have been reported. The first one is the production of heterocyte glycolipids by a PKS cluster [88]. When a vegetative cell differentiates into a future heterocyte, the nascent cell only becomes active in fixing nitrogen when a layer of glycolipids covers it. This layer prevents oxygen, produced by adjacent cells, to enter into the heterocyte and to inhibit the nitrogenase. This mechanism must be tightly regulated and programmed because heterocytes can only survive 3 to 4 days before being replaced by another heterocyte resulting from the differentiation of a vegetative cell. The second example is from the product of a PKS cluster, the nostopeptolide that governs the cellular differentiation of a symbiotic *Nostoc* [89,90]. This example also indicates a clear scheduling of the production of the natural product at the time needed by the producing organism. Finally, the last example of the need of the producer to produce certain molecules at a certain time can be seen through the study of the toxic-bloom forming *Microcystis aeruginosa* PCC 7806. With an ingenious culture system consisting of two compartments separated by a filter, which allows compounds but not cells to pass, Briand and his collaborators demonstrated that this strain produced certain NPs in the medium only by sharing it with another strain of *Microcystis* such as PCC 9432 or *Planktothrix agardhii* PCC 7805 [100,101]. Thus, *Microcystis aeruginosa* PCC 7806 produces these NPs when it detects another cyanobacterium in its environment. This allelopathic research deserves further study because it illustrates a very controlled production of these metabolites beyond the genetic potential of the producer. It can also lead to the discovery of so-called cryptic NPs.

## 6. Concluding Remarks

In conclusion, the Pasteur Cultures of Cyanobacteria collection has been a living biobank and a research tool since its creation at the Institut Pasteur. The status of these strains has allowed research in the global scientific community. Within the framework of cyanobacterial toxins, the strains of the PCC collection led to the discovery of cyanotoxins and NPs. While several cyanotoxins were already structurally known, the pure strains maintained at the Institut Pasteur for 52 years have made it possible to discover the genetic origins of these toxins, intriguing enzymes, even unprecedented chemistry, and certain bioactivities. The genomics of the strains of the PCC collection highlights the wide diversity of NPs that we are still fully investigating.

## Figures and Tables

**Figure 1 toxins-15-00388-f001:**
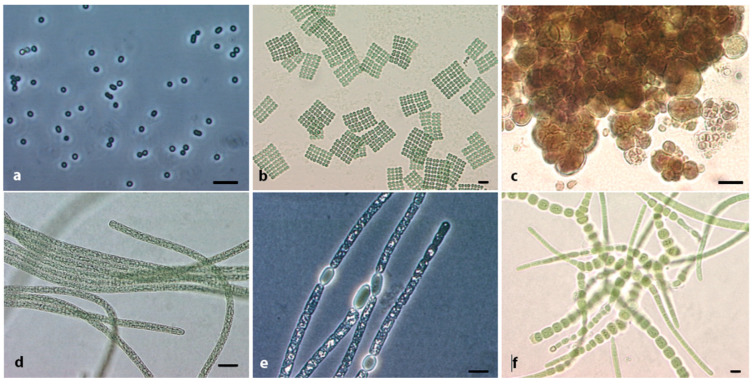
Morphologies of cyanobacteria. Unicellular morphotypes are on the top row: from tiny single cells of *Cyanobium* sp. PCC 7001 in dark field (**a**), to flat square colonies of *Merismopedia*, from an environmental sample (**b**) to the baeocytous former *Stanieria* sp. PCC 7301 (**c**). Filamentous morphotypes on the bottom row: from *Planktothrix agardhii* PCC 10110 with bright gas vacuoles (**d**) to *Aphanizomenon flos aquae* PCC 7905 in dark field with gas vacuoles and barrel-shaped heterocyte (**e**), to *Fischerella* sp. in the late stage of purification (**f**). All scale bars represent 5 µm.

**Figure 2 toxins-15-00388-f002:**
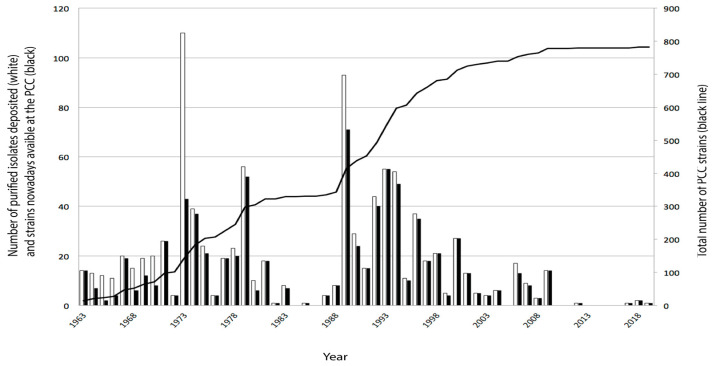
Number of axenic monoclonal cyanobacterial strains of the PCC obtained over the years 1963–2019 (in white), cultures still preserved today (2023, in black), and total number of PCC strains (black line and second vertical axis).

## Data Availability

Not applicable.

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
