# Peer review of "Cyanotoxins and Other Bioactive Compounds from the Pasteur Cultures of Cyanobacteria (PCC)"

_toxins, 2023, doi:10.3390/toxins15060388_

Round 1

Reviewer 1 Report

Due to the combined/interactive effects of eutrophication and changes in climate, cyanobacterial blooms and associated cyanotoxin contamination have been increasing worldwide. The frequency, intensity, and duration of blooms are expected to increase further due to the eutrophication and warming trends. These blooms pose risks to ecosystems and health of humans. This paper presented a timely review on cyanotoxin and various bioactive compounds from the aspect of the Pasteur Cultures of Cyanobacteria (PCC). It is generally well written. Some comments are raised however as follows:

1.  From the title, it is not easy to understand what the central topic is addressed in this paper. It would be better to include some more words specify the research area related to "Cyanotoxins and other bioactive compounds". 

2. Abstract: it is not clear what have been addressed in this review. History, achievement, challenge, perspective related to the research on the cyanotoxins and other bioactive compounds from the Pasteur Cultures of Cyanobacteria (PCC)?

3. Introduction: L. 33-34, Refs are needed here to support the reported toxic blooms and their potential impacts. For example, Huisman, J., Codd, G.A., Paerl, H.W., Ibelings, B.W., Verspagen, J.M.H., and Visser, P.M. (2018). Cyanobacterial blooms. Nature Reviews Microbiology 16, 471–483.

4. Section 4 Natural products of cyanobacteria, from toxins to novel compounds, L 167-180, some more recent papers have been published concerning the potential impacts of cyanotoxin in African continent. For example, the paper of “Wang, H., Xu, C., Liu, Y., et al. (2021). From unusual suspect to serial killer: Cyanotoxins boosted by climate change may jeopardize megafauna. The Innovation. 2(2),100092.” about the cyanotoxin-poisoning-related massive death of African elephants; and also the one of “Xu Zhao, Ying Liu, Yu-Ming Guo, Chi Xu, …, Hans W. Paerl, Erik Jeppesen, Ping Xie. 2023. Meta-analysis reveals cyanotoxins risk across African inland waters. Journal of Hazardous Materials, 451(1):131160

5. Concluding remarks: it would be better to include some conclusion on the general achievement, challenge, and some comments of perspective.

It is generally well written with regards to English language.

Author Response

Thanks for the comments but the paragraph on the eutrophication and changes in climate, cyanobacterial blooms and associated cyanotoxin contamination is not the central topic of this review.

1) The goal of this requested review was dedicated to L. Pasteur (Chemist, microbiologist and founder of our institution) in celebration of the bicentenary of his birth in a Special issue of Toxins. We were asked to describe what is done in the frame of toxins and related compounds with the collection of Cyanobacteria maintained at the Institut Pasteur, specifically with the history of this collection. Thus it is not about 'Cyanotoxins and other bioactive compounds' but on 'Cyanotoxins and other bioactive compounds from the Pasteur Cultures of Cyanobacteria (PCC)'.

2) Exactly, History, achievement, challenge, perspective related to the research on the cyanotoxins and other bioactive compounds from the Pasteur Cultures of Cyanobacteria (PCC)are what we have addressed in this review.

3) Thank you reference added.

4) we agree we have now added information on the African wildlife threaten by cyanobacterial proliferations and toxins. thank you for this suggestion.

5) We already concluded on the achievement made with the PCC strain from the 70's to now: meaning useful to the entire scientific community, also on challenge to find new natural products from these strains, and our perspectives to be even more clear: The genomics of the strains of the collection PCC highlight the wide diversity of NPs, that we are still fully investigating. We could have expand to a full page, but it would be just a dilution of our goal.

Reviewer 2 Report

Manuscript ID: toxins-2404728

Cyanotoxins and other bioactive compounds from the Pasteur Cultures of Cyanobacteria

In the frame of the bicentenary of the birth of Louis Pasteur, this manuscript focuses on the cyanotoxins and other natural products from cyanobacteria at the collection Pasteur Cultures of Cyanobacteria. It is well written and informative, describing the early history of cyanobacteria, cyanotoxins and other compounds at PCC as well as the contributions of early scientists. This review also summarizes the main and important contributions by the PCC researchers to the cyanobacteria and natural products field. Although the focus is the main results obtained by the PPC researchers, the authors also take into account and cite works from other groups.

The manuscript presents an interesting topic and highlights the importance of this cyanobacteria collection, which has been a useful resource for the scientific community, namely within the cyanobacteria one.

 Minor comments:

Line 32-34: Cyanobacteria produces cyanotoxins but also other compounds with promising activities (e.g. anticancer, antibiotic). Probably in the introduction, this general information should be mentioned.

Line 36: … and compound structure elucidation and…

Line 38:  Suggestion: …discovered.... little is known about them

Line 37-38: Abbreviate Natural products (NPs)

Line 39: Suggestion: delete “about the cyanotoxins and other bioactive compounds from the collection Pasteur Cultures od Cyanobacteria” (it is already said that the review is about that).

Line 159: change natural products to NPS (it is abbreviated before). Do the same along the text

Line 166: suggestion: change the title to “Natural products of cyanobacteria – from toxins to novel compounds”

Line 199: delete “natural products” and leave the abbreviation NPs. Since the beginning that natural products is referred along the text. It doesn’t make sense to abbreviate only here.

Line 225: correct multidisciplinary

Line 286: change diminution to decrease 

Author Response

Thank you very much for your comments and thorough reading of our manuscript, we have now taken most of your suggestions:

Line 32-34 although promising, the main problem is about how were conducted these bioactivity tests, and we really don't want to go there in this review. We have added this information in the section '5 from molecules to bioactivities'.

lines 36, 38, 39, 166, 225 and 286: well spotted modified accordingly, thank you

We have now indicated NPs after the first occurrence and change this along the text, thank you for noticing.

Thank you for your support.

Reviewer 3 Report

The manuscript has been nice presentation about the current topic, However, some major changes required :

1. English correction, grammar, and syntax deeply need to be taken care of.

2. one segment is required details about the interaction of the cyanotoxins with contaminants like antibiotics, nanoparticles etc.

3. A detailed schematic section will be better for the explanation.

4. recent citations are required .

The English grammer, syntax, spelling, and syntax need to be checked through out the manuscript.

Author Response

thank you for your comments,

Most reviewers were ok with our text, but I sent it to an English speaker to make it better.

You suggested that one segment is required details about the interaction of the cyanotoxins with contaminants like antibiotics, nanoparticles etc. But this completely out of the scope asked to us and we are not expert in this aspect, nor knowing any work done with PCC strains or their toxins with contaminants.

You asked: a detailed schematic section will be better for the explanation. Thus I wonder an explanation of what? this review, in honor of L. Pasteur, the founder of our institution for its 200 birthday anniversary, is describing what is the PCC, history and work done with these strains in general and what we make with the PCC strains related to cyanotoxins and other bioactive compounds in particular.

 About requirement of recent citations: our citations are comprised from year 1875 to 2022, with 29 citations out of the 98 references from the last 5 years. What more recent citations one can expect ?

Reviewer 4 Report

Comments

An interesting and up-to-date review is presented. Undoubtedly, it will be of interest to the readers of the Toxins journal.

There are some comments aimed at improving the perception of the drawings presented in the review, as well as the understanding of the text.

1) Figure 1. It is better to enter letter designations (a,b, c..) for each photo of cyanobacteria placed in the figure and make appropriate explanations in the description of the figure (in the legend to the figure).

2) Figure 2. X-axis - the years are indicated in very small font. The Authors can make a markup and put it, for example, an inscription for every second year.

3) A reference to the corresponding study on line 223 is needed.

4) Probably, the review could provide data on the number of sequenced genomes representing cyanobacteria strains from the Pasteur collection (this is not a requirement, just a wish).

The English language of the article is clear. Authors can show the text to a native English-speaking colleague for verification.

Author Response

Thank you for your review, comments that helped to improve our manuscript.

1) Figure 1. now fixed with letter, you are right it is better.

2) Figure 2. X-axis again thank you, right now we indicate every 5 years and it is more clear.

3) The reference of the sentence line 223 is the whole table 1. we found it simpler to write a table than to add 33 references in a text.

4) About your wish on providing data on the number of sequenced genomes representing cyanobacteria strains from the Pasteur collection: We could but the pressure is so strong when the communities knows that we will release genomes of the PCC (as the 54 genomes we released in the Shih et al, 2013, too much solicitations prior publication!) that we are avoid saying it in advance. For you only, we already sequenced half of the PCC strains.

Thank you for your support

Round 2

Reviewer 3 Report

Author has described and addressed all the comments . It can be recommended for acceptance now. however, the way of responding to the comments is very rude and unscientific. Authors group should work on this. And the editorial committee should look over the things for future aspects.